# A scoping review of information provided within degenerative cervical myelopathy education resources: Towards enhancing shared decision making

Rishi Umeria[1], Oliver Mowforth[1,2], Ben Grodzinski[1], Zahabiya Karimi[2], Iwan Sadler[2], Helen Wood[2], Irina Sangeorzan[2], Petrea Fagan[3], Rory Murphy[4], Angus McNair[5], Benjamin Davies[1,2]*

1 Department of Clinical Neurosciences, Division of Neurosurgery, University of Cambridge, Cambridge, United Kingdom, 2 Myelopathy.org, Cambridge, United Kingdom, 3 School of Clinical Medicine, University of Cambridge, Cambridge, United Kingdom, 4 Department of Neurosurgery, Barrow Neurological Institute, Phoenix, Arizona, United States of America, 5 Centre for Surgical Research, Bristol Medical School: Population Health Sciences, University of Bristol, Bristol, United Kingdom

* bd375@cam.ac.uk

**Data Availability Statement:** All relevant data are within the manuscript and its Supporting Information files.

## Abstract

### Background

Degenerative cervical myelopathy (DCM) is a chronic neurological condition estimated to affect 1 in 50 adults. Due to its diverse impact, trajectory and management options, patient-centred care and shared decision making are essential. In this scoping review, we aim to explore whether information needs in DCM are currently being met in available DCM educational resources. This forms part of a larger Myelopathy.org project to promote shared decision making in DCM.

### Methods

A search was completed encompassing MEDLINE, Embase and grey literature. Resources relevant to DCM were compiled for analysis. Resources were grouped into 5 information types: scientific literature, videos, organisations, health education websites and patient information leaflets. Resources were then further arranged into a hierarchical framework of domains and subdomains, formed through inductive analysis. Frequency statistics were employed to capture relative popularity as a surrogate marker of potential significance.

### Results

Of 2674 resources, 150 information resources addressing DCM were identified: 115 scientific literature resources, 28 videos, 5 resources from health organisations and 2 resources from health education websites. Surgical management was the domain with the largest number of resources (66.7%, 100/150). The domain with the second largest number of resources was clinical presentation and natural history (28.7%, 43/150). Most resources (83.3%, 125/150) were designed for professionals. A minority (11.3% 17/150) were written for a lay audience or for a combined audience (3.3%, 5/150).

**Funding:** The author(s) received no specific funding for this work.

**Competing interests:** The authors have declared that no competing interests exist.

## Conclusion

Educational resources for DCM are largely directed at professionals and focus on surgical management. This is at odds with the needs of stakeholders in a lifelong condition that is often managed without surgery, highlighting an unmet educational need.

## Introduction

In his address to the Royal College of Surgeons in 1923, Rudyard Kipling described words as "the most powerful drug used by mankind". This metaphor reflects how good communication can have profound effects on clinical decision making [1, 2], the well-being of patients and can improve patient experience of any consultation [3], even if no treatment is offered [4].

Communication is particularly important in chronic conditions such as degenerative cervical myelopathy (DCM) [5, 6], which is estimated to affect 2% of adults [7]. DCM arises when degenerative changes in the cervical spine precipitate a mechanical stress injury of the spinal cord [8]. This leads to progressive disability, including loss of manual dexterity, imbalance, sensory disturbance and pain. Surgery to decompress the spinal cord is the only evidence-based management [9–11]. However, surgery carries risks, may not be required in mild and stable cases, and despite surgery, fewer than 5% of people make a full recovery. Most people with DCM consequently require long-term support for a range of disabilities.

DCM management decisions are therefore complex, individualised [12] and dependent on contextual and technical factors. In particular, the variable and unpredictable trajectory [8], the role of disease surveillance [13] and the importance of managing recovery expectations [14] mandate high-quality communication. Patients must be well informed and have the opportunity to apply and utilise the information that is communicated [15, 16].

Achieving this is challenging. Many factors contribute to effective communication [2], not least selection and prioritisation of information [17], which is prone to bias leading to variability in the quantity or quality of information exchanged [18, 19]. This is pertinent to DCM where complex care pathways involve numerous different professional disciplines [20, 21] often with limited and outdated knowledge of the condition [22, 23]. In addition, the selection of information for resources such as leaflets and websites is commonly driven by the perceptions of healthcare professionals [24].

Providing information effectively for individual consumers requires a person-centred approach, which can be enhanced by the use of communication tools [25]. The objective is to provide the right information, in the right quantity, at the right time, in a manner that can be understood by the individual [26]. This approach is recognised widely in clinical guidelines, including the NHS Person-Centred Approaches Framework [27], the General Medical Council guidance on decision making and consent [28] and is endorsed by the NHS England Personalised Care Institute [29]. Myelopathy.org is specific DCM charity, which was founded to address the challenges faced by the community of people living with DCM [5, 30].

The objective of this scoping review was to undertake a structured exploration of the information provided by current DCM educational resources. The aim is to facilitate development of additional tools to support communication in DCM, forming part of a larger Myelopathy. org initiative to promote shared decision making in DCM.

## Methods

### Resource type categorisation

DCM educational resources were categorised into 5 types: scientific literature, videos, health organisations, health education websites and hospital information leaflets. Consensus on

**Table 1. Search strategy for organisations, health education websites and hospital patient information leaflets.**

| Method | Tool | Search term | Additional information |
|---|---|---|---|
| 1 | Select webpage on patient or professional resources from navigation menu | Cervical myelopathy | N/A |
| 2 | Searchbar on website | Cervical myelopathy | Search performed once for a given website |
| 3 | Find in page function (Ctrl + F) | Cervical myelopathy | Search repeated for each webpage page on a given website |

Hierarchical search strategy to identify educational content on DCM from health organisation websites, health education websites and hospital patient information leaflets. Method 1 was employed first. Method 2 was employed if no information on DCM was found using method 1. Method 3 was used employed if no information was found using methods 1 and 2. If no information found using all 3 methods, the resources was excluded.

resource types was reached by the authorship group, which included people with DCM. Health seeking behaviour of the public and health information provision from healthcare professionals was considered when developing the categorisation system [31]. For example, healthcare professionals frequently distribute patient information leaflets (PILs) [32, 33] and the public increasingly access health education and health organisation websites and online videos [34–36].

## Search strategy

A comprehensive search strategy was developed and refined for each of 7 key resource domains (S1 Appendix). For the health organisation and health education websites, a hierarchical search strategy was employed to identify DCM educational content (Table 1).

Overall inclusion and exclusion criteria were created (Table 2) and adapted for each specific resource type (S2 Appendix). Educational content exclusively addressing degenerative cervical myelopathy was sought. Content covering cervical myelopathy of non-degenerative aetiology was excluded.

## Resource types

**1. Scientific literature.** High sensitivity DCM search filters were used to search the databases EMBASE [37] and MEDLINE [38] (S3 Appendix). The PROSPERO database, a prospectively maintained register of planned systematic reviews, was also searched using the term "myelopathy". Searches were performed on 6$^{th}$ May 2020. To identify currently relevant information, searches were limited to the past 3 years for narrative reviews and the past 20 years for systematic reviews. These article types were selected because narrative reviews typically aim to provide educational content and systematic reviews focus on specific clinical questions.

**Table 2. Overall criteria for screening for DCM educational resources.**

| Inclusion Criteria | Exclusion Criteria |
|---|---|
| English language | Heterogenous populations (not exclusively DCM or CSM +/- OPLL) |
| Educational tool | Cervical myelopathy of non-degenerative aetiology |
| Degenerative cervical myelopathy OR Cervical spondylotic myelopathy +/- OPLL | Cervical radiculopathy |

Overall inclusion and exclusion criteria for screening resources to identify those with educational DCM content. These criteria were applied to screen resources of from all 5 key resource types: scientific literature, videos, health organisations, health education website and hospital patient information leaflets. The aim was to identify public-facing resources that contained educational content on DCM. Specific inclusion and exclusion criteria were then adapted for each resource type (S2 Appendix).

**2. Videos.**   Google is the most popular search engine, accounting for 93% of all internet searches [39]. A search of 'cervical myelopathy' was run under the *videos* subsection of Google on 23rd April 2020 to identify educational videos on DCM. In total, 54% of videos (15/28) returned in the search were from YouTube and 46% (13/28) of videos were from other multi-media websites.

**3. Organisations.**   A comprehensive global list of organisations with potentially relevant educational content on DCM were identified by the AO Spine RECODE-DCM ENVIROS-CAN project. This project is collaborative research effort, aiming to increase DCM research efficiency [40]. The ENVIROSCAN is diverse list of organisations–charities, hospitals, universities and professional academic bodies–either involved in, or with the potential to be involved in, DCM research. A hierarchical search strategy was used to search 737 organisation websites (Table 1).

**4. Health education websites.**   Alexa Top Sites, part of Amazon Web Services, was used to identify the most popular websites under the category of health, subcategory education, based on website traffic on 6th May 2020 [41]. The top sites feature of Alexa uses a traffic rank algorithm based on relative reach and page views over the previous three months by Alexa users to determine the highest reaching websites. The hierarchical search strategy was used to search the top 50 health education websites.

**5. Hospital patient information leaflets.**   A list of the hospitals recorded as specialised providers of complex spinal surgery from Appendix C of the spinal services report (S4 Appendix) [42], captures all hospitals in the United Kingdom offering surgical treatment for DCM [11, 20]. Our hierarchical search strategy was used to search 40 hospital websites for information on DCM provided by the hospital.

## Data extraction

Educational information was extracted from included resources by one author (RU) at two separate time points, to ensure all educational content was extracted.

## Resource domain development

Extracted information was then categorised inductively by two authors (RU and BMD) into 7 key information domains: aetiology, pathophysiology and epidemiology; clinical presentation and natural history; diagnosis and monitoring progression; surgical management; non-surgical management; predicting outcomes; assessing research and developing guidelines (Table 3). Domains were developed independently by assessing all information resources and iteratively refined until they were applicable across all content, comprehensively covered all key concepts

**Table 3. Domain categorisation system for DCM educational resources.**

| Domain number | Domain name |
|---|---|
| 1 | Aetiology, pathophysiology and epidemiology |
| 2 | Clinical presentation and natural history |
| 3 | Diagnosis and monitoring progression |
| 4 | Surgical management |
| 5 | Non-surgical management |
| 6 | Predicting outcomes |
| 7 | Assessing research and developing guidelines |

Seven key domains were identified to categorise educational content on DCM in to.

and consensus between authors was reached. One author (BMD) had prior knowledge about DCM, whilst the other author (RU) did not, meaning the final categorisation system reflected a reconciliation between a knowledge-driven and an unbiased approach. Agreed information domains were further divided into a framework of information subdomains to enable content analysis of information from different resources using descriptive statistics (Table 4). Additional criteria were used to determine if each resource was targeted at a lay or professional audience or both (Table 5). A specific distinction of whether a resource was for patients or other lay stakeholders was not attempted.

## Results

Of the 2674 resources screened, 150 DCM information resources were identified: 115 from the scientific literature, 28 videos, 5 from organisations and 2 from health education websites (Fig 1). No hospital information leaflets were identified. Reporting for this review followed the Preferred Reporting Items for Systematic Reviews and Meta-analyses (PRISMA) checklist (S5 Appendix).

The most common domain addressed was surgical management, with 67% (100/150) of resources. The least common domain was assessing research and developing guidelines, with 7% (10/150) of resources (Fig 2). Approximately 11% (16/150) of resources covered nonsurgical management. The majority of resources (86%, 129/150) were designed for a professional audience; a minority were targeted at a lay audience (11%, 16/150) or a joint professional and lay audience (3%, 5/150).

### Domain 1—Aetiology, pathophysiology and epidemiology (33 resources)

Key themes identified were that the disease process is poorly understood, however there appears to be consensus that both static and dynamic injury contribute to chronic cervical cord compression. Most resources in this domain (82%, 27/33) discussed DCM pathophysiology, including ischaemia, disruption of the spinal cord-blood barrier, inflammation and apoptosis.

A total of 12% (4/33) of resources in this domain consider the role of genetic factors in DCM, such as the fact that only small percentage of individuals with radiological evidence of cervical cord compression have clinical features of DCM, may represent a genetic susceptibility to DCM.

In total, 33% (11/33) of resources addressed epidemiology. Scientific literature resources in particular, often report DCM as the leading cause of spinal cord dysfunction in adults worldwide, however, frequently lack precise estimates of prevalence and incidence. Common reasons cited include the lack of a standardised definition of DCM, difficulty in diagnosis, and a lack of large-scale observational studies. Myelopathy.org reports that up to 5% of people over the age of 40 may have DCM [43]. Age is widely cited as an important risk factor, with degenerative changes being more common with age and most people with DCM being diagnosed in their 50s. An important consideration is how the incidence of DCM may increase with a globally ageing population [44].

### Domain 2—Clinical presentation and natural history (43 resources)

A total of 79% (34/43) of resources in this domain described the symptoms of DCM, with fewer reporting on signs (37%, 16/43). Clinical presentation was often covered thoroughly describing the classical presentation of neck pain or stiffness, poor manual dexterity, clumsiness, paraesthesia, gait dysfunction and bladder and bowel dysfunction. Over 75% (33/43) of resources in this domain reported on the natural history of DCM, often commenting on the

**Table 4. Summary of DCM educational resources categorisation.**

| Title | Code | Narrative Reviews (16) | | Systematic Reviews (99) | | Videos (28) | | Organisations (5) | | Health Education Websites (2) | |
|---|---|---|---|---|---|---|---|---|---|---|---|
| Aetiology, pathophysiology and epidemiology | 1 | 12 | 75% | 6 | 6% | 11 | 39% | 2 | 40% | 2 | 100% |
| Aetiology | 1a | 2 | 13% | 4 | 4% | 1 | 4% | 0 | 0% | 0 | 0% |
| Pathophysiology | 1b | 11 | 69% | 2 | 2% | 10 | 36% | 2 | 40% | 2 | 100% |
| Epidemiology | 1c | 11 | 69% | 0 | 0% | 0 | 0% | 0 | 0% | 0 | 0% |
| Clinical presentation and natural history | 2 | 11 | 69% | 6 | 6% | 20 | 71% | 4 | 80% | 2 | 100% |
| Symptoms | 2a | 10 | 63% | 0 | 0% | 18 | 64% | 4 | 80% | 2 | 100% |
| Signs | 2b | 8 | 50% | 1 | 1% | 7 | 25% | 0 | 0% | 0 | 0% |
| Natural history | 2c | 10 | 63% | 5 | 5% | 12 | 43% | 4 | 80% | 2 | 100% |
| Diagnosis and monitoring progression | 3 | 12 | 75% | 11 | 11% | 7 | 25% | 3 | 60% | 1 | 50% |
| Diagnosis | 3ai | 10 | 63% | 8 | 8% | 7 | 25% | 3 | 60% | 1 | 50% |
| Monitoring progression | 3aii | 4 | 25% | 2 | 2% | 0 | 0% | 0 | 0% | 0 | 0% |
| Clinical assessment | 3bi | 10 | 63% | 6 | 6% | 7 | 25% | 3 | 60% | 1 | 50% |
| Radiological assessment | 3bii | 10 | 63% | 1 | 1% | 5 | 18% | 0 | 0% | 0 | 0% |
| Electrophysiological assessment | 3biii | 7 | 44% | 1 | 1% | 1 | 4% | 1 | 20% | 0 | 0% |
| Biomarker assessment | 3biv | 3 | 19% | 1 | 1% | 0 | 0% | 0 | 0% | 0 | 0% |
| Surgical management | 4 | 11 | 69% | 61 | 62% | 22 | 79% | 4 | 80% | 2 | 100% |
| Surgical approach decision | 4a | 8 | 50% | 0 | 0% | 6 | 21% | 0 | 0% | 0 | 0% |
| Surgical procedure—anterior only | 4bi | 0 | 0% | 14 | 14% | 1 | 4% | 0 | 0% | 0 | 0% |
| Surgical procedure—posterior only | 4bii | 0 | 0% | 19 | 19% | 1 | 4% | 0 | 0% | 0 | 0% |
| Surgical procedure—both | 4biii | 8 | 50% | 25 | 25% | 9 | 32% | 1 | 20% | 1 | 50% |
| Surgical procedure—neither | 4biv | 3 | 19% | 0 | 0% | 11 | 39% | 3 | 60% | 1 | 50% |
| Surgical outcomes | 4c | 10 | 63% | 61 | 62% | 6 | 21% | 1 | 20% | 0 | 0% |
| Non-surgical management | 5 | 6 | 38% | 4 | 4% | 3 | 11% | 2 | 40% | 1 | 50% |
| Physiotherapy | 5ai | 3 | 19% | 4 | 4% | 2 | 7% | 1 | 20% | 1 | 50% |
| Medications | 5aii | 6 | 38% | 2 | 2% | 1 | 4% | 1 | 20% | 1 | 50% |
| Cervical traction | 5aiii | 5 | 31% | 3 | 3% | 0 | 0% | 1 | 20% | 0 | 0% |
| Cervical bracing | 5aiv | 5 | 31% | 1 | 1% | 1 | 4% | 1 | 20% | 1 | 50% |
| Bedrest | 5av | 3 | 19% | 0 | 0% | 0 | 0% | 0 | 0% | 0 | 0% |
| Avoidance of risky activities/environment | 5avi | 2 | 13% | 0 | 0% | 0 | 0% | 0 | 0% | 0 | 0% |
| Orthoses | 5avii | 1 | 6% | 2 | 2% | 0 | 0% | 0 | 0% | 0 | 0% |
| No specific interventions (unspecified) | 5aviii | 1 | 6% | 1 | 1% | 1 | 4% | 0 | 0% | 0 | 0% |
| Non-surgical outcomes | 5b | 5 | 31% | 4 | 4% | 0 | 0% | 0 | 0% | 0 | 0% |
| Predicting outcomes | 6 | 7 | 44% | 19 | 19% | 1 | 4% | 0 | 0% | 0 | 0% |
| Clinical predictors | 6ai | 6 | 38% | 9 | 9% | 1 | 4% | 0 | 0% | 0 | 0% |
| Imaging predictors | 6aii | 7 | 44% | 10 | 10% | 0 | 0% | 0 | 0% | 0 | 0% |
| Non-specified predictors | 6aiii | 0 | 0% | 2 | 2% | 0 | 0% | 0 | 0% | 0 | 0% |
| Surgical outcomes | 6bi | 6 | 38% | 15 | 15% | 1 | 4% | 0 | 0% | 0 | 0% |
| Non-surgical outcomes | 6bii | 1 | 6% | 1 | 1% | 0 | 0% | 0 | 0% | 0 | 0% |
| Natural disease course Outcomes | 6biii | 1 | 6% | 9 | 9% | 0 | 0% | 0 | 0% | 0 | 0% |
| Assessing research and developing guidelines | 7 | 3 | 19% | 5 | 5% | 1 | 4% | 1 | 20% | 0 | 0% |
| Future directions | 7ai | 3 | 19% | 0 | 0% | 0 | 0% | 0 | 0% | 0 | 0% |
| Reporting outcome measures | 7aii | 0 | 0% | 3 | 3% | 0 | 0% | 0 | 0% | 0 | 0% |
| Reporting trends | 7aiii | 0 | 0% | 1 | 1% | 0 | 0% | 0 | 0% | 0 | 0% |
| Developing guidelines | 7b | 0 | 0% | 1 | 1% | 1 | 4% | 1 | 20% | 0 | 0% |

For each resource type the number of resources in the domain and subdomain is recorded and the percentage of resources containing information on the domain or subdomain, of all the resources of that type. A single resource can contain information on more than one domain or subdomain within a domain.

**Table 5. Criteria for target audience determination for DCM educational resources.**

| Patient Criteria | Professional Criteria |
|---|---|
| Information delivered in style for individual from non-healthcare background, including use of simple language (avoiding medical jargon) | Information delivered in style for individual from healthcare background, including use of medical jargon, clinical management applications, research study applications |
| Pre-determined audience by source nature e.g. patient information leaflet (PIL) | Pre-determined audience by source nature e.g. scientific literature |
| Source content providing information for an individual with DCM on symptoms to look out for, what to expect through interactions with healthcare professionals for accessing treatment pathway, how they can adapt to living with their condition | Information delivered to audience with a presumed knowledge of anatomy, physiology or pathology of the spine and spinal cord |
| | Source content providing information on how to take a history, perform a physical examination, investigations to order or approach to management of an individual with DCM |

Criteria to determine the target audience of each resource: patients, professionals or a combined audience Resources that met components of both the patient and professional criteria were recorded as having a combined audience.

variability and unpredictability. A frequent statistic cited by several resources, is that 20% to 62% of people with DCM experience clinical deterioration (defined by a change of at least one point in the modified Japanese Orthopaedic Association (mJOA) score over a 3-to-6-year follow-up period [45].

## Domain 3—Diagnosis and monitoring progression (34 resources)

Diagnosis was discussed by 85% (29/34) of resources in this domain. A key theme addressed was the need for specialist involvement, including the importance for doctors in primary healthcare to refer a person with suspected DCM to a spinal surgeon, who may fall under the remit of neurosurgery or orthopaedics, for a specialist assessment.

Radiological assessment and clinical assessment were discussed by 47% (16/34) and 79% (27/34) of resources respectively. The combination of MRI of the cervical spine alongside clinical signs and symptoms was an important point frequently made regarding diagnosis. There is also consensus that DCM is difficult to diagnose and that there is poor awareness of the condition among the general public and non-specialist healthcare professionals.

The topic of monitoring progression, reported on by 18% (6/34) resources, involved commentary that different resources refer to different tools to assess functional impairment in DCM. This has led to inconsistencies in assessing outcomes. The mJOA scale and Nurick scale are commonly used DCM-specific indices, yet both have poor sensitivity [46–48].

## Domain 4—Surgical management (100 resources)

This domain was covered by 67% (100/150) of all resources. Key themes were that surgery aims primarily to decompress the spinal cord and aims secondarily to stabilise the spinal column. Common anterior and posterior surgical approaches, such as anterior cervical discectomy and fusion and laminectomy were frequently discussed. A total of 15% (15/100) of resources in this domain reported on anterior approach only, 20% (20/100) on posterior only, 44% (44/100) on both anterior and posterior, and 18% (18/100) discussed surgical procedure more generally. A key theme covered by 14% (14/100) of resources, including both videos and scientific literature, were factors that determine surgical approach. Systematic reviews were the most common resource type (61%, 61/100) to report on surgical management.

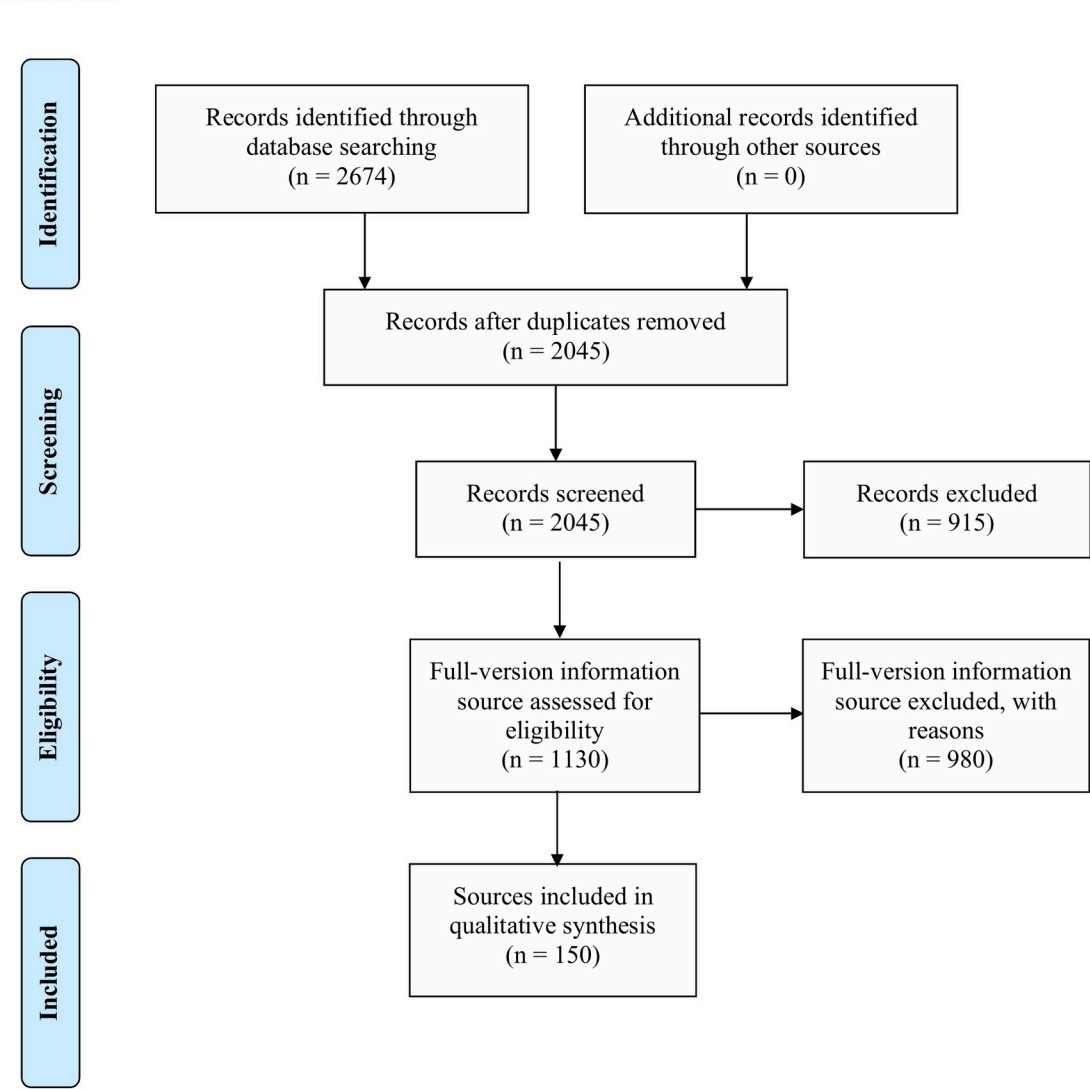

**Fig 1. PRISMA (Preferred Reporting Items for Systematic Reviews and Metanalyses) flow diagram of search strategy.** The process of identifying the educational resources that met inclusion criteria from databases illustrated as a flow diagram.

Surgical outcomes, including efficacy and complications of surgery, were discussed by 78% (78/100) of resources in this domain. Another frequently discussed theme was the role of surgery in stopping the progression of the disease and preventing further neurological decline. A small number of scientific literature resources cited recently growing evidence that surgery may improve neurological function. Resources frequently refer to clinical guidelines published by AO Spine in 2017, which strongly recommend surgery for moderate to severe DCM and recommend a structured trial of rehabilitation or surgery for mild DCM [11]. Surgical complications were another key theme. For the anterior approach this included dysphagia, damage to local structures, pseudoarthrosis and for the posterior approach it included C5 palsy and surgical site infection.

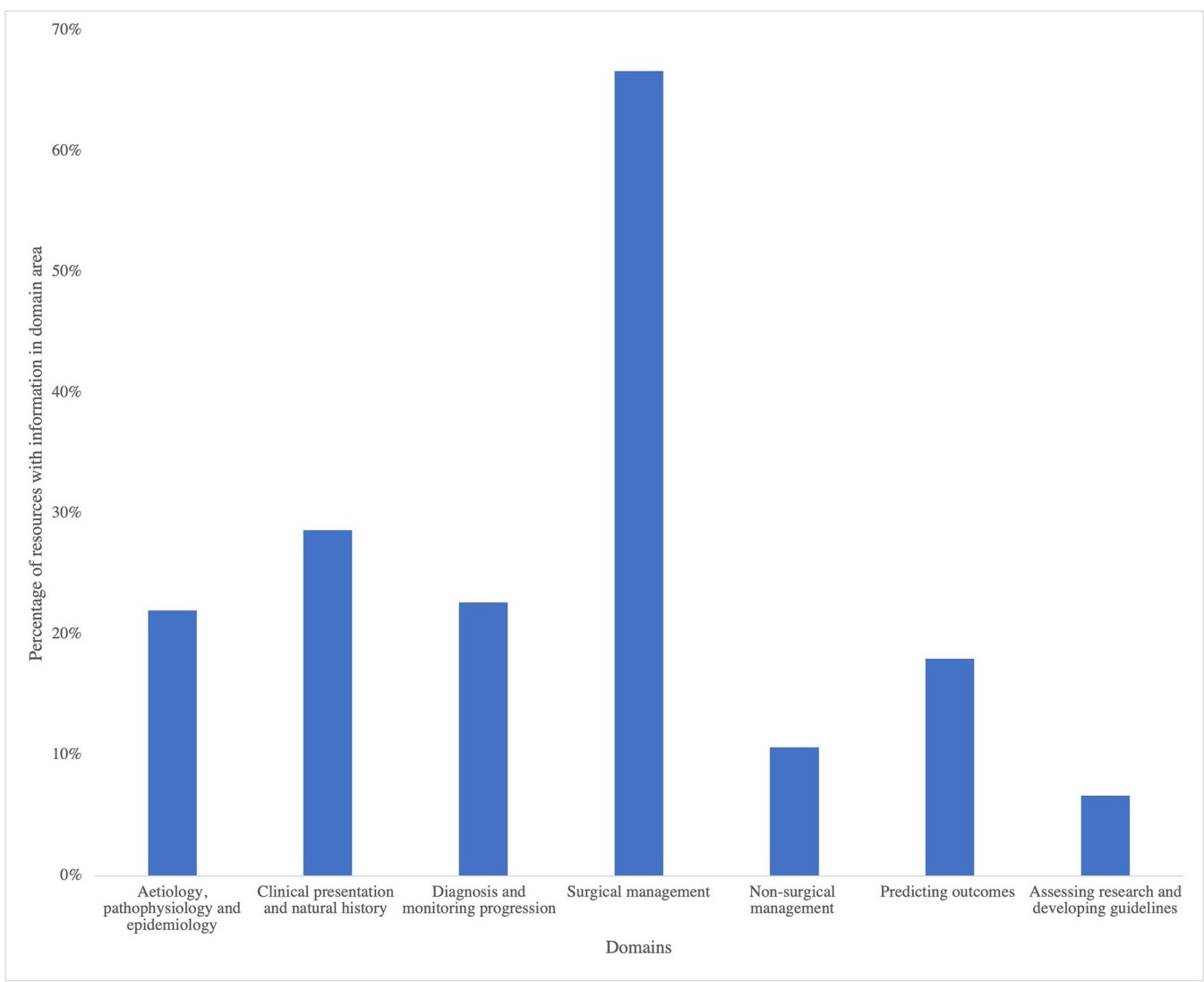

**Fig 2. Percentage of educational resources on each of the 7 DCM domains.** The most common domain was surgical management (67%). The least common domain was assessing research and developing guidelines (7%).

### Domain 5—Non-surgical management (16 resources)

There was considerably lower coverage of the nonsurgical management domain compared to the surgical management domain. A total of 63% (10/16) of nonsurgical management resources were from the scientific literature. Resources described the consensus that nonoperative management does not provide definitive treatment for spinal cord compression but may have a role in symptomatic management for mild DCM. Nonoperative management discussed included physiotherapy (69%, 11/16), medication (69%, 11/16), cervical traction (56%, 9/16), orthoses (19%, 3/16), bedrest (19%, 3/16) and avoidance of high-risk activities (13%, 2/16). In total, 56% (9/16) of resources in this domain discussed outcomes of non-surgical management, all of which were from the scientific literature. The lack of high-quality evidence to support the role of non-surgical management was commonly discussed.

### Domain 6—Predicting outcomes (27 resources)

Predicting outcomes was covered by 18% (27/150) of resources. All resources were from the scientific literature, except for one video, discussing modelling outcome prediction in DCM. A key theme was the use of clinical data and imaging data to predict outcomes, discussed by 59% (16/27) and 63% (17/27) of resources, respectively. Literature resources discussed factors that may determine if a person is likely to benefit from surgery; surgical outcomes were discussed by 81% (22/27) of resources in this domain, while outcomes from non-operative management were discussed by 7% (2/27) of resources. Commonly cited factors associated with poorer surgical outcomes include older age, longer symptom duration, and worse preoperative disease severity [49]. Predicting outcomes in the natural disease course was addressed by 37% (10/27) of resources in this domain. This appears to be an important theme for people with DCM, as predicting outcomes can manage a person's expectations [14] and lead to expeditious referral for people with predictors of rapid deterioration.

### Domain 7—Assessing research and developing guidelines (10 resources)

In total, 7% (10/150) of resources provided information on assessing research and developing guidelines. An important theme was inconsistency in terminology and outcomes reducing the efficiency of research, which was covered by 30% (3/10) of resources in this domain. Future research to further the field was discussed by 30% (3/10) of resources. Advances in imaging, including quantitative MRI, and the use of biomarkers are two key upcoming areas. Another theme is addressing knowledge gaps about DCM, for example with respect to clinical practice guidelines the optimal treatment strategy for mild DCM remains unknown [11].

## Discussion

Following a comprehensive search, 150 DCM educational resources were analysed. The majority were targeted at professionals, rather than a lay audience. Information provision largely focused on surgical management and to a lesser extent clinical presentation.

### Dominance of the scientific literature and surgery

More than three-quarters (77%, 115/150) of all resources identified came from the scientific literature, this included systematic reviews (66%, 99/150) and narrative reviews (11%, 16/150). The two domains with the highest proportion of resources coming from the scientific literature were domain 6 on predicting outcomes (96%, 26/27) and domain 7 on assessing research and developing guidelines (80%, 8/10). Of all 115 resources from the scientific literature, the two most common domains covered were domain 4 on surgical management (63%, 72/115) and domain 6 (23%, 26/115). As educational content from the scientific literature is directed at healthcare professionals, it is not surprising that there is a focus on surgery, the main treatment modality for DCM. The content is especially important for informing guidelines and making evidence-based decisions on the surgical approach for spinal decompression [50].

Of all 72 resources from the scientific literature that covered surgical management, 71 (99%) addressed outcomes of surgery, compared to 27% (6/22) of videos, 25% (1/4) of resources from organisation and 0% (0/2) of resources from health education websites. Only 4% (3/72) of literature resources on surgical management discussed surgical procedures in general, non-technical terms, whereas non-literature resources were more likely to: 50% (11/22) of videos, 75% (3/4) of organisations, 50% (1/2) health education websites discussed surgery more generally.

For the 28 video resources, surgical management was the most common domain (79%, 22/28) and the second most common domain was clinical presentation and natural history (71%, 20/28). Videos were targeted at the general public and professionals with equal frequency. Understanding how to recognise the disease and explaining how it is managed were common notable features of videos. The small number of organisations and health education websites identified, limited depth analysis of these resource types and highlights the paucity of DCM educational resources.

## Lack of information on DCM

The total of 150 educational resources identified within the parameters of this review, highlights the limited educational resources available on DCM, especially for a lay audience (11%, 16/150). Possible factors that may be contributing include the variable terminology used for the condition and the fact that there are many aspects of DCM that are not well understood. These factors may create barriers for development of educational resources since answers to certain questions about the condition simply do not exist. The AO Spine RECODE-DCM project has established global research priorities to address knowledge gaps in the field and this will help coordinate the research effort to tackle this issue [40].

All domains, except surgical management, were covered by less than 30% of the educational resources. Domain 1 coverage was particularly low (22%, 33/150), which may be due to poor understanding of DCM aetiological factors, pathophysiological mechanisms, and lack of large epidemiological datasets to inform accurate estimates of incidence and prevalence. Furthermore, domain 2 coverage may be low (29%, 43/150) due to the variability in initial presentation and disease progression. Moreover, low domain 3 coverage (23%, 34/150) may be because DCM is difficult to diagnose, especially early in the disease course, where symptoms may be non-specific, making appropriate educational resources challenging to produce.

Explaining monitoring of progression in resources can be a challenge when there is a lack of consensus on what tools should be used. Very poor coverage of domain 5 (11%, 16/150), is likely due to the lack of evidence supporting non-surgical management in DCM, making it difficult to justify the inclusion in educational resources, especially when focusing on evidence-based management. Domain 6 coverage being low (18%, 27/150), might again be explained by the paucity of evidence for outcome prediction, in particular identifying factors that will determine which patients will be most likely to benefit from surgery. Domain 7 (7%, 10/150) had the lowest coverage. This is partly due to this being almost exclusively the remit of scientific literature. Assessing patterns and commenting on the progress of research in the field is challenging when the field of DCM research is still relatively small [51].

Resources from outside the scientific literature were in the minority (23%, 35/150). This aligns with the paucity of resources targeted at a lay audience, who are generally less likely to obtain their information on DCM from the scientific literature [31, 52]. As the DCM research field grows, there is potential for information from the scientific literature to be filtered into other types of resources that are more likely to reach the lay audience.

## Generalisability

A focus on surgery is consistent with the wider scientific literature [53]. This is understandable given that surgery is currently the only disease modifying therapy [54]. However experience from AO Spine RECODE DCM [40] and related projects has demonstrated that this will overlook critical perspectives [55]. For example, a survey by Myelopathy.org identified that pain was the number one recovery priority for people living with DCM [56], yet research to date has rarely measured it [57, 58]. Furthermore, a focus on surgery does not address the large

proportion of people with DCM who are currently managed non-operatively [59, 60], the many different professionals involved in this process, [20] and the long-term implications that persist following surgery [61, 62].

This review therefore indicates that education gaps exist within DCM, due to its paucity but also lack of breadth. However, what this review does not characterise is the information need of people living with DCM. This is the objective of a qualitative study being undertaken in parallel by Myelopathy.org.

With this information, the aim is to develop solutions to support personalised information exchange in DCM. One approach may be the formation of core information sets (CIS)—refined lists of critical topics to be discussed, for example in supporting informed consent for surgery [25]. CIS are formed using a multi-stakeholder consensus process, which involves initially gathering information using literature reviews and interviews, before distilling this into a core list of information. CIS are therefore designed to be a starting point to help professionals, patients and their carers ensure key information is considered. They are intended to be personalised during each consultation and have been pioneered in surgical oncology [63–65].

CIS likely have value far beyond surgical consent. In DCM, patients often share common or stereotyped transitions points in their care, such as obtaining diagnosis or preparing for surgical treatment. This makes a series of CIS a potentially effective adjuvant to improve communication and outcomes in DCM. The information domains identified in this review may align well with this. Furthermore, Myelopathy.org has had success using this methodological approach in AO Spine RECODE DCM [20, 40, 65–67]. The definitive next steps will need to be considered in the context of the information needs identified amongst people living with DCM.

## Limitations

Resources were selected pragmatically using a multi-stakeholder perspective and searches were only conducted online. Relevant educational resources, including physical content such as printed information leaflets and local resources not freely available online may therefore not have been captured. The consequence of this, on the conclusions drawn in the article, is unknown. Patients are now known to use the internet as their predominant source of healthcare information over healthcare professionals [68, 69]. Nonetheless, it is difficult to say with certainty the impact printed PILs have on patient behaviour as it depends on the context of the clinical situation and the invasiveness of the intervention [32, 70].

The literature search was performed in May 2020, identifying pre-COVID-19 pandemic data on educational resources. Resources produced during the pandemic will not have been captured. However, the pre-pandemic resources are more likely to be consistent with addressing the research question and help orienting clinical research and the formation of CIS once the COVID-19 pandemic is over.

Furthermore, our inclusion criteria stipulated information sources should be specifically focused on DCM. DCM information may have been grouped with other medical conditions such as cervical radiculopathy, as well as being placed in the broader category of non-traumatic spinal cord injury [71, 72]. This appears a particular issue with hospital patient information leaflets; 10 leaflets providing generic information for cervical surgery were identified as surgical procedures used to treat DCM are common to many other diseases.

However, this is unlikely to be a limitation for DCM CIS. Firstly, owing to a poor awareness of DCM amongst the general public and the general medical community [73, 74], the identification of relatively few printed educational resources was expected. This knowledge gap led to the foundation of Myelopathy.org, which receives an international audience. Moreover, the

search covered all resources recognised as relevant to patients seeking health information [31, 75]. Finally, this review will be supplemented by qualitative studies, such as interviews, to identify relevant information for a CIS. This is likely to be of greater and specific relevance to produce patient centred CIS [25].

## Conclusion

There are few dedicated educational resources for people with DCM or the general public. The majority of education material is found within the scientific literature for a professional audience. Key areas of focus included surgical management; clinical presentation and natural history; non-surgical management; and diagnosis and monitoring for progression. Aetiology, pathophysiology, epidemiology, predicting outcomes and developing guidelines were also addressed by professionally orientated resources. This information will be used to inform a larger initiative by Myelopathy.org to support patient centred care in DCM.

## Supporting information

**S1 Appendix. Search strategy identifying educational resources in scientific literature, videos, organisations, health education websites and hospital patient information leaflets.**
(DOCX)

**S2 Appendix. Inclusion and exclusion criteria for educational resources in scientific literature, videos, organisations, health education websites and hospital patient information leaflets.**
(DOCX)

**S3 Appendix. DCM search filters for scientific literature databases: EMBASE and MEDLINE.**
(DOCX)

**S4 Appendix. List of UK hospitals offering complex spinal surgery services–taken from Spinal Services GIRFT Programme National Specialty Report Appendix C.**
(DOCX)

**S5 Appendix. PRISMA checklist.**
(DOC)

## Acknowledgments

BMD is supported by a NIHR Clinical Doctoral Research Fellowship. ODM is supported by an Academic Clinical Fellowship.

The views expressed in this publication are those of the authors and not necessarily those of the NHS, the National Institute for Health Research or the Department of Health and Social Care.

## Author Contributions

**Conceptualization:** Benjamin Davies.

**Methodology:** Ben Grodzinski.

**Project administration:** Zahabiya Karimi.

**Writing – original draft:** Rishi Umeria, Benjamin Davies.

**Writing – review & editing:** Rishi Umeria, Oliver Mowforth, Iwan Sadler, Helen Wood, Irina Sangeorzan, Petrea Fagan, Rory Murphy, Angus McNair, Benjamin Davies.

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
