## [Decision Letter · Decision Letter 0]

25 Feb 2022

PONE-D-22-02227A scoping review of information provided within degenerative cervical myelopathy education resources: towards enhancing shared decision makingPLOS ONE

Dear Dr. Umeria,

Thank you for submitting your manuscript to PLOS ONE. After careful consideration, we feel that it has merit but does not fully meet PLOS ONE’s publication criteria as it currently stands. Therefore, we invite you to submit a revised version of the manuscript that addresses the points raised during the review process.

We look forward to receiving your revised manuscript.

Kind regards,

Michael G. Fehlings

Academic Editor

PLOS ONE

Journal Requirements:

Additional Editor Comments:

In addition to the points outlined by the reviewers, the authors need to address the perceived conflict of interest related to myelopathy.org (which was flagged by a reviewer in private comments to the editor).

Reviewers' comments:

Reviewer's Responses to Questions

**Comments to the Author**

1. Is the manuscript technically sound, and do the data support the conclusions?

Reviewer #1: Yes

Reviewer #2: Yes

2. Has the statistical analysis been performed appropriately and rigorously? 

Reviewer #1: N/A

Reviewer #2: N/A

3. Have the authors made all data underlying the findings in their manuscript fully available?

Reviewer #1: Yes

Reviewer #2: Yes

4. Is the manuscript presented in an intelligible fashion and written in standard English?

Reviewer #1: Yes

Reviewer #2: No

5. Review Comments to the Author

Reviewer #1: Thanks for the submission.

Line 271: It was unanticipated that no hospital issued information was identified. Could the authors hypothesise why that was, given the high incidence of DCM?

Line 332: 1st mention of mJOA without defining the acronym (is is defined in line 339)

Line 340: Please reference this statement

Line 522: Suspect that this is a very significant weakness of this work. Printed information remains the mainstay of patients informing in secondary care in many centres. Could the authors comment any evidence as to how significantly this could influence conclusions if included?

Reviewer #2: This represents a much needed study on the unmet needs of current provision of educational support to DCM stakeholders, namely patients, carers and healthcare professionals.

I have few comments and two references to suggest:

1) The methodology clearly states that the search was conducted in May 2020. I would suggest mentioning this aspect in the Limitation of the Study section: some documents may have been missed as a result of this choice, however it should also me noted that the search identified pre-pandemic data that might be more consistent with the research question and help orienting clinical research once COVID-19 will be over.

2) I suppose you have run your search for videos on both Google and Youtube, you should state that the first is more generic the second more specific for this resource type.

3) There are two articles that I would suggest adding to the reference list, both aimed at a surgical and AHCP audience, one on surgical decision making and one on the future challenges of DCM management, they would certainly represent a good contribution for the discussion on domains 4 and 7:

- Kato S, Ganau M, Fehlings MG. Surgical decision-making in degenerative cervical myelopathy - Anterior versus posterior approach. J Clin Neurosci. 2018 Dec;58:7-12. doi: 10.1016/j.jocn.2018.08.046.

- Ganau M, Holly LT, Mizuno J, Fehlings MG. Future Directions and New Technologies for the Management of Degenerative Cervical Myelopathy. Neurosurg Clin N Am. 2018 Jan;29(1):185-193. doi: 10.1016/j.nec.2017.09.006.

Once again, many thanks for the opportunity to review such a nice scoping review.

6. PLOS authors have the option to publish the peer review history of their article (what does this mean?). If published, this will include your full peer review and any attached files.

Reviewer #1: No

Reviewer #2: **Yes: **Mario Ganau, Consultant Neurosurgeon - Oxford University Hospitals

---

## [Author Response · Author response to Decision Letter 0]

31 Mar 2022

UNIVERSITY OF CAMBRIDGE

Division of Neurosurgery

Department of Clinical Neurosciences

Addenbrooke’s Hospital

University of Cambridge

CB2 0QQ, United Kingdom

 Email bd375@cam.ac.uk

Professor Michael G. Fehlings

Academic Editor

PLOS ONE 

Re: Our manuscript titled “A scoping review of information provided within degenerative cervical myelopathy education resources: towards enhancing shared decision making” by R. Umeria, O. Mowforth, B. Grodzinski, Z. Karimi, I. Sadler, H. Wood, I. Sangeorzan, P. Fagan, R. Murphy, A. McNair, B. Davies

Dear Professor Fehlings,

We thank you and the reviewers for reviewing our manuscript and for offering us the opportunity to revise it.

We have made all suggested revisions and enclose our revised manuscript. Please find below a point-for-point response to all the editor’s and reviewers’ suggestions.

We hope you agreed that our manuscript is now much improved and suitable for publication in PLOS ONE.

Yours sincerely, 

Mr Benjamin Davies

Founder and Research Director, Myelopathy.org (DCM Charity)

NIHR Doctoral Research Fellow, University of Cambridge

Specialist Registrar Neurosurgery, Cambridge University Hospital

Point for point response

J= journal comment 

R = reviewer comment

E= additional editor comment

A = author response

Journal Requirements 

J: When submitting your revision, we need you to address these additional requirements.

A: Thank you for providing the links to the style templates to check our manuscript meets the style requirements of the journal. We have reviewed the templates and made the appropriate changes to meet PLOS ONE’s style requirements: 

1. Qualifications have been removed in the author by-line 

2. Author affiliation details have been amended 

3. Physical address details for the corresponding author have been removed 

4. The referencing style has been modified so all references come before punctuation marks 

5. The table legends have been moved to underneath the table 

6. The font size has been standardised for all tables 

Additional Editor

E: In addition to the points outlined by the reviewers, the authors need to address the perceived conflict of interest related to myelopathy.org (which was flagged by a reviewer in private comments to the editor).

A: We thank the reviewer for raising the important topic of conflict of interest. Myelopathy.org is a registered charitable organisation, governed by the Charity Commission for England and Wales (charity number 1178673). Myelopathy.org exists to serve the interests of the DCM patient community through its charitable objectives. We do not consider that any of the relationships between authors of this article and the charity, introduced any conflicts of interest. For example, we cannot envisage personal gain of any of the authors from this article due to their affiliation with myelopathy.org. The resource would be freely available, as part of the publication within PLOS One.

Reviewer: 1

R: Thanks for the submission.

A: We thank the reviewer for reviewing our manuscript. 

R: Line 271: It was unanticipated that no hospital issued information was identified. Could the authors hypothesise why that was, given the high incidence of DCM?

A: We thank reviewer 1 for highlighting this point. We have ensured that this is addressed in the limitations. As we only searched hospital websites for their information leaflets, we acknowledge that this may have missed printed leaflets, or leaflets only available on local intranets. It is worth noting that a number of leaflets were identified, but as they were generic for the procedure (e.g. ACDF) rather than the condition of DCM, these were excluded. Given most healthcare information is now sought online, and hospital websites do typically make available their information leaflets, we think this is unlikely overall to have influenced our findings.

R: Line 332: 1st mention of mJOA without defining the acronym (is defined in line 339). 

A: Thank you. We have corrected this so that mJOA acronym is defined in full the first time it appears in the text.

R: Line 340: Please reference this statement.

A: Thank you. We have added references to support this point.

R: Line 522: Suspect that this is a very significant weakness of this work. Printed information remains the mainstay of patients informing in secondary care in many centres. Could the authors comment any evidence as to how significantly this could influence conclusions if included?

A: We acknowledge in the limitations section that our study design did not capture printed leaflets, which was intentional in our methodological design. Unfortunately, it was not feasible within the scope of this review to systematically survey the written information provided to patients by all hospitals across the world. Nonetheless, the internet is now the leading source of healthcare information for patients ahead of healthcare providers[1]. Looking towards the future, printed information sheets of heterogeneous content and quality provided by individual centres will almost certainly be superseded by high quality, universally available online educational patient materials, thus we decided to focus our efforts here for this review.

Additionally, the impact patient information leaflets have on patient behaviour is known to depend on the context of the clinical situation and the invasiveness of the intervention [2,3]. It is therefore difficult to determine the exact size of the impact of having potentially missed printed patient information leaflets on the conclusions drawn in this article.

Reviewer 2

R: This represents a much needed study on the unmet needs of current provision of educational support to DCM stakeholders, namely patients, carers and healthcare professionals.

A: We thank reviewer 2 for reviewing our manuscript.

R: I have few comments and two references to suggest: 1) The methodology clearly states that the search was conducted in May 2020. I would suggest mentioning this aspect in the Limitation of the Study section: some documents may have been missed as a result of this choice, however it should also me noted that the search identified pre-pandemic data that might be more consistent with the research question and help orienting clinical research once COVID-19 will be over.

A: Thank you for raising this point, we have now added commentary in the limitations subsection of the discussion section, highlighting that pre-pandemic data is most likely to be closely aligned to answering the research question in this article. 

R: 2) I suppose you have run your search for videos on both Google and YouTube, you should state that the first is more generic the second more specific for this resource type.

A: We thank reviewer 2 for highlighting this point. Our video search was run for ‘cervical myelopathy’ under the videos subsection of the Google search engine, which yielded results both for videos that were from YouTube (15 out of 28) and videos that were not from YouTube (13 out of 28). We have added further discussion to the methodology section to make this clear.

R: 3) There are two articles that I would suggest adding to the reference list, both aimed at a surgical and AHCP audience, one on surgical decision making and one on the future challenges of DCM management, they would certainly represent a good contribution for the discussion on domains 4 and 7:

- Kato S, Ganau M, Fehlings MG. Surgical decision-making in degenerative cervical myelopathy - Anterior versus posterior approach. J Clin Neurosci. 2018 Dec;58:7-12. doi: 10.1016/j.jocn.2018.08.046.

- Ganau M, Holly LT, Mizuno J, Fehlings MG. Future Directions and New Technologies for the Management of Degenerative Cervical Myelopathy. Neurosurg Clin N Am. 2018 Jan;29(1):185-193. doi: 10.1016/j.nec.2017.09.006.

A: Thank you for drawing our attention to these important references, which we have now cited at the appropriate places within the discussion section.

R: Once again, many thanks for the opportunity to review such a nice scoping review.

References 

1. Swoboda, Christine M., et al. “Odds of Talking to Healthcare Providers as the Initial Source of Healthcare Information: Updated Cross-Sectional Results from the Health Information National Trends Survey (HINTS).” BMC Family Practice, vol. 19, no. 1, 29 Aug. 2018, 10.1186/s12875-018-0805-7.

2. Sustersic M, Gauchet A, Foote A, Bosson J-L. How best to use and evaluate Patient Information Leaflets given during a consultation: a systematic review of literature reviews. Health Expectations [Internet]. 2016 Sep 26;20(4):531–42. Available from: https://onlinelibrary.wiley.com/doi/full/10.1111/hex.12487

3. 70. Kenny T. A PIL for every ill? Patient information leaflets (PILs): a review of past, present and future use. Family Practice. 1998 Oct 1;15(5):471–9.

---

## [Decision Letter · Decision Letter 1]

26 Apr 2022

A scoping review of information provided within degenerative cervical myelopathy education resources: towards enhancing shared decision making

PONE-D-22-02227R1

Dear Dr. Umeria,

We’re pleased to inform you that your manuscript has been judged scientifically suitable for publication and will be formally accepted for publication once it meets all outstanding technical requirements.

Kind regards,

Michael G. Fehlings

Academic Editor

PLOS ONE

Additional Editor Comments (optional):

Reviewers' comments:

Reviewer's Responses to Questions

**Comments to the Author**

1. If the authors have adequately addressed your comments raised in a previous round of review and you feel that this manuscript is now acceptable for publication, you may indicate that here to bypass the “Comments to the Author” section, enter your conflict of interest statement in the “Confidential to Editor” section, and submit your "Accept" recommendation.

Reviewer #1: All comments have been addressed

Reviewer #2: All comments have been addressed

2. Is the manuscript technically sound, and do the data support the conclusions?

Reviewer #1: Yes

Reviewer #2: Yes

3. Has the statistical analysis been performed appropriately and rigorously? 

Reviewer #1: Yes

Reviewer #2: Yes

4. Have the authors made all data underlying the findings in their manuscript fully available?

Reviewer #1: Yes

Reviewer #2: Yes

5. Is the manuscript presented in an intelligible fashion and written in standard English?

Reviewer #1: Yes

Reviewer #2: Yes

6. Review Comments to the Author

Reviewer #1: Thank you for addressing the comments. I do think that paper information remains an important component of information to the patient population, but I appreciate the explanations given.

Reviewer #2: The authors have appropriately answered all comments from reviewers/editors. I would like to congratulate them on such a nice study.

I am pleased to endorse their manuscript for publication in PLOS ONE in the present form.

7. PLOS authors have the option to publish the peer review history of their article (what does this mean?). If published, this will include your full peer review and any attached files.

Reviewer #1: No

Reviewer #2: **Yes: **Mr Mario Ganau, Comsultant Neurosurgeon, Oxford University Hospitals, UK

---

## [Editor Report · Acceptance letter]

11 May 2022

PONE-D-22-02227R1 

A scoping review of information provided within degenerative cervical myelopathy education resources: towards enhancing shared decision making 

Dear Dr. Umeria:

I'm pleased to inform you that your manuscript has been deemed suitable for publication in PLOS ONE. Congratulations! Your manuscript is now with our production department. 

Kind regards, 

on behalf of

Dr. Michael G. Fehlings 

Academic Editor

PLOS ONE